# Surface Modification-Dominated Space-Charge Behaviors of LDPE Films: A Role of Charge Injection Barriers

**DOI:** 10.3390/ma15176095

**Published:** 2022-09-02

**Authors:** Yuanwei Zhu, Haopeng Chen, Yu Chen, Guanghao Qu, Guanghao Lu, Daomin Min, Yongjie Nie, Shengtao Li

**Affiliations:** 1State Key Laboratory of Electrical Insulation and Power Equipment, Frontier Institute of Science and Technology, Xi’an Jiaotong University, Xi’an 710049, China; 2Electric Power Research Institute, Yunnan Power Gird Co., Ltd., Kunming 650217, China

**Keywords:** space charge, high voltage, LDPE, surface oxidation, charge injection, insulating polymer

## Abstract

Gradually increasing power transmission voltage requires an improved high-voltage capability of polymeric insulating materials. Surface modification emerges as an easily accessible approach in enhancing breakdown and flashover performances due to the widely acknowledged modification of space-charge behaviors. However, as oxidation and fluorination essentially react within a limited depth of 2 μm underneath polymer surfaces, the nature of such bulk space-charge modulation remains a controversial issue, and further investigation is needed to realize enhancement of insulating performance. In this work, the surface oxidation-dependent space-charge accumulation in LDPE film was found to be dominated by an electrode/polymer interfacial barrier, but not by the generation of bulk charge traps. Through quantitative investigation of space-charge distributions along with induced electric field distortion, the functions of surface oxidation on the interfacial barrier of a typical dielectric polymer, LDPE, is discussed and linked to space-charge behaviors. As the mechanism of surface modification on space-charge behaviors is herein proposed, space-charge accumulation can be effectively modified by selecting an appropriate surface modification method, which consequentially benefits breakdown and flashover performances of polymeric insulating films for high-voltage applications.

## 1. Introduction

Non-conjugated polymers such as polyethylene, polypropylene and epoxy resin (EP) serve as the main electrical insulating component in power equipment [1,2,3,4,5,6,7,8]. Due to the gradually increasing power transmission voltage, advanced polymer insulators are required to sustain long-term extremely high electrical strengths [5,9,10,11]. Space charges, injected from a metal electrode, inevitably accumulate inside the bulk of insulating polymers, which are considered as the main reason for inner electrical field distortion, leading to accelerated deterioration and breakdown of the polymeric insulating materials [12,13,14,15].

Based on the recently developed high-resolution space-charge detection methods such as pulsed electroacoustics (PEA) and pressure wave propagation (PWP), the existence, profile and functions of space charges on insulation discharge have been realized [16,17,18,19]. DC high voltage induces continuous space-charge injection and accumulation, which trigger increased electric field strength in the middle bulk, leading to breakdown of the insulating film [14,15,17,20,21]. Differently, due to the periodic alternated polarities of AC voltage, large quantities of space charges accumulate near film/electrode interfaces, giving rise to great electric field distortion, which leads to the decreased breakdown strength as compared to that under DC voltages [15]. Although most of the references reported decreased breakdown strength by the introduction of space charges, surface charges contribute to improved flashover performance for most of the insulating polymers [22,23,24]. Recent investigations on electron beam-irradiated EP indicate increased density of deep-trapping sites which suppress the multiplication of secondary electrons, and consequently results in enhanced flashover performance [25,26,27].

Since space charge shows the double-edged sword effect on high voltage discharges, its regulation strategy has become a research hotspot, among which surface modification is an emerging approach. Investigations on surface-fluorinated polyimide and EP show generation of shallow traps with increased surface conductivity, which improves flashover performance [28,29]. Similar enhancement is achieved with surface oxidation on low-density polyethylene (LDPE) and high-density polyethylene (HDPE) [30], and there exists opposite arguments about surface fluorination, which simultaneously inhibits space-charge injection and accumulation in LDPE [31,32,33].

Thus, divergence still exists on the surface modification-dependent space-charge behaviors of polymer insulating materials; thus, further investigation and understanding are needed. Furthermore, the surface treatment of polymer insulators will unavoidably lead to modified interfacial contact between metal electrodes and insulating film, resulting in varied space-charge injecting behaviors which largely modulate the absolute quantities of space charges [34]. However, such major factors, namely, the injection barrier between the electrode and dielectric has not been systematically investigated, which restricts further understanding on the surface treatment-dependent space-charge behaviors of polymer dielectrics.

Here, based on surface-oxidized LDPE films, the charge injection barrier is investigated and linked to the accumulation of space charges. With the assistance of numerical simulations, the injection barrier-dominated space-charge profile of a model dielectric polymer LDPE is investigated, providing in-depth understanding of surface treatment-dependent space-charge behaviors.

## 2. Materials and Methods

**Materials:** LDPE with the nameplate of LE4147 was purchased from Borealis (Vienna, Austria), which exhibits a nominal density of 0.922 g/cm^3^, and a dielectric constant of 2.3. For preparing the LDPE film, the LDPE pellet was first heated at 60 °C, and then thermal pressed at 140 °C, 15 MPa for 15 min. After that, the LDPE film was cooled down to ambient room temperature. To conduct surface oxidation, the prepared LDPE film was placed into a reactor with 0.04 MPa, 110 mg/L ozone atmosphere. At each duration of 1 h, 2 h, 4 h and 6 h, the oxidized LDPE film was taken out and labeled.

**Characterization:** X-ray photoelectron spectroscopy (XPS) was conducted with a Thermo Fisher ESCALAB Xi+. Thermally stimulated current (TSC) was performed with a Novocontrol Concept 80 system. During testing, the LDPE film with sputtered gold electrodes was polarized at 80 °C, 2 kV/mm for 30 min, and then rapidly cooled down to −100 °C with a temperature ramping rate of −30 °C/min. After that, a short-circuited depolarization process was conducted for 3 min, and the temperature was re-increased to 90 °C at 3 °C/min for obtaining the TSC. The total trap amount and trap level of the LDPE film was obtained through a fitting process [20,35,36]. A PEA test was conducted under 30 kV/mm, with polarization and depolarization durations of 1200 s and 300 s, at room temperature.

**Simulations:** For first-principle simulations, an LDPE molecular chain with the polarization of 10 ethylene monomers was established. Before calculating the energy band structure, the configuration optimization of the molecular chain was first conducted, and the ORCA software package [37] was used to optimize the lowest energy configuration under the B3LYP function [38,39], with DFT-D3-quantified dispersion corrections. For the numerical simulations of space-charge behaviors, the space-charge injection and transport processes were described by the Schottky thermionic emission, charge transport and charge trapping–detrapping dynamic equations, as reported in our early investigations [40,41]. The space-charge-induced electric field distortion along the film thickness direction was described by Poisson’s equation [42].

## 3. Results and Discussion

### 3.1. Characterisation of Surface Oxidation

XPS was first applied in characterizing the surface element within a depth of ca. 10 nm of the LDPE film after ozone treatment. As shown in Figure 1a, three peaks at the binding energy of 282 eV, 528 eV and 974 eV were observed, which represented C1s, O1s and O auger electrons, respectively. The intensity of these three peaks increased with the prolonged oxidation duration, indicating that the surface oxidation treatment had successfully introduced O atoms onto the LDPE film.

More detailed information is reported in Figure 1b,d. In Figure 1b of the C1s photoelectron spectroscopy, despite the barely changed intensity of C1s peak at 282 eV, a small peak was observed at 286.5 eV, which corresponds to C=O. A left shift of the C=O peak was observed in inset of Figure 1b, indicating the raising binding energy of C=O with the increased oxidation duration. Figure 1c is the O1s spectrum, showing a gradually enhanced peak intensity with the prolonged oxidation duration, which further indicates the growing absolute quantity of introduced O atoms, proving the surface modification of LDPE by ozone treatment. Figure 1d is the O auger peak, the trend of which shows a similar increase with that of O1s in Figure 1c.

Generally, the intensity of the photoelectron peaks in XPS spectrum represents the concentration of O element in the surface layer of the LDPE film. A higher intensity of the peak corresponds to a larger O intensity. Based on the XPS results in Figure 1, ozone treatment introduced O atoms onto the surface of the LDPE film through forming a C=O bond, and the concentration of O element increased with the continuous extension of the oxidation treatment duration. A supporting data of Fourier transform infrared spectroscopy (FTIR) with a gradually emerging C=O peak at 1710 cm^−1^ also suggests the grafting of O atoms onto the LDPE surface, as reported in Ref. [26]. Thus, the surface modification of the LDPE film was successfully achieved through UV-ozone treatment.

### 3.2. Characterisation of Charge Traps

Surface treatment of dielectric films could lead to varied space-charge performance. Here, the space-charge characteristics of LDPE film under ozone treatment are investigated through thermally stimulated current (TSC) tests, the result of which are reported in Figure 2.

Three peaks were observed and labeled as peak α, peak β and peak γ, arranged from high temperature to low temperature, respectively. Earlier investigation indicated that the generation mechanism of peak γ was dipole polarization, and peak α was generated by the accumulation of space charges [35,43]. In Figure 2a, the TSC of peak α was apparently larger than those of peak β and peak γ, which indicates the minor contribution to the total quantity of space charges from peak β and peak γ. As the TSC of peak α continuously increased with oxidation duration, it is thus verified that the space-charge accumulation tended to be more violent after ozone treatment of the LDPE film.

In order to quantitatively investigate the variation in the space-charge behaviors of the LDPE after ozone treatment, the trap level and trap amount of peak α were extracted from the TSC curves, by applying [35,43],
(1)$$I_{TSC}\left(T\right)=\frac{p}{\tau }exp\left[-\frac{E_{T}}{kT}-\frac{1}{\beta \tau }\int_{T_{0}}^{T}exp\left(-\frac{E_{T}}{kT}\right)dT\right]$$
where *I_TSC_* is the TSC in A; *T* is the test temperature in K; *p* is the polarization intensity in C/m^2^; *τ* is the relaxation time in s; *E_T_* is the trap level in eV; *k* is the Boltzmann constant; and *β* is the temperature ramping rate in K/s.

The variations in trap level and trap amount with the changes in oxidation duration is shown in Figure 2b and Table 1. The as-prepared LDPE film exhibited a trap level of 0.66 eV, and the surface oxidation lead to a tremendously increased trap level of 0.76 eV after 1 h ozone treatment. However, further prolonging of the oxidation treatment resulted in a decreased trap energy towards 0.63 eV under 6 h treatment. Differently, the trap amount showed a continuous increase with ozone treatment, from 1.44 nC of as-prepared LDPE film to 9.00 nC of 6 h treated film, indicating an increased quantity of space charges. This increasing trend of space charges is in accordance with that of the PEA results as reported in ref. [26], which suggests an increase from 2.5 nC of as-prepared film to 9.4 nC of 6 h ozone treated film.

As is widely acknowledged, surface oxidation could only occur within several micrometers of the polymer film, and the chemical structure in the bulk of the film stayed unchanged. TSC represents the space-charge behaviors of the film bulk. In other words, the quantity of space charges along the film depth direction is integrally accounted for [36]. However, the TSC results in Figure 2 show extremely large variations with oxidation duration, which indicates that the charge trap information of the whole film was changed by surface oxidation.

Earlier investigation suggests that surface shallow traps are generated by oxidation treatment, which has been well proved by isothermal surface potential decay (ISPD) experiments. These shallow traps on the film surface facilitate space-charge injection from the electrodes, which results in more accumulated charges in the bulk of the film, and consequently leads to an enlarged TSC.

PEA experiments were conducted, and the result was compared with that of TSC. PEA is a common method for characterizing the space-charge behaviors of a dielectric material and the PEA results of the LDPE film after surface oxidation are shown in Figure 3.

In Figure 3a, for as-prepared LDPE film, the HOMO charge density near the cathode reached ca. −30 C/m^3^, while that near the anode reached ca. 8 C/m^3^. Comparatively in Figure 3b,c, for LDPE films after surface oxidation treatment, a large increase was observed in space-charge density. For instance, in Figure 3b, the negative charge density near the cathode reached ca. −50 C/m^3^, and the positive charge density near the anode reached ca. 20 C/m^3^, which were nearly double those of the as-prepared LDPE film. These results indicate that the quantity of space charges in the bulk of the LDPE film was tremendously increased after surface oxidation, and a more detailed result is reported in Figure 3d.

In Figure 3d, for the calculated total amount of space charges, both the TSC and PEA results show greatly enhanced space charge accumulation by surface oxidation, while the ISPD result shows a barely changed trap amount (data obtained from [44]). As ISPD applies non-electrode contact for space-charge injection, such an interesting result indicates that surface oxidation does not naturally change the bulk trap features of the dielectric films. However, surface treatment inevitably modulates the charge injection barrier between the electrode and the film surface. Thus, as both TSC and PEA apply electrode contact for space-charge injection, the quantities of accumulated charges increased tremendously after surface oxidation treatment.

### 3.3. Characterisation of Charge Injection Barrier

The above-mentioned discussion suggests that the changed trap parameters extracted from TSC and PEA do not actually represent the bulk trap information, but are mainly related to the changes in charge injection and accumulation, controlled by the surface trap conditions. This consideration is also proved by the barely changed trap energy alongside the greatly varied trap amount. Earlier ISPD results also suggest that surface oxidation introduces shallow traps on the surface of dielectric films, resulting in much more violent space-charge injection [26]. This promoted charge injection by surface shallow traps could essentially be attributed to the varied electrode/polymer interfacial barriers, which has been further investigated by first-principle molecular simulations.

In respect of the measurement of the charge injection barrier, we have also considered experimentally measuring the HOMO level of the surface-oxidized LDPE films by ultraviolet photo-electron spectroscopy (UPS) and obtaining the charge injection barrier by subtractive calculation with the work function of the metal electrode. However, UPS testing requires vertical electrical conduction along the film thickness direction, and thus LDPE films with excellent electrical insulating performance can hardly be tested. After careful reading of the literature, it is still an extremely hard issue to experimentally measure the energy band structure of such dielectric polymers, and thus, a molecular simulation is carried out.

The energy level structures of C_20_H_42_ (LDPE with a polymerization index of 10) and C_20_H_40_O (LDPE with a one grafted O atom) were first investigated, as demonstrated in Figure 4. C_20_H_42_ exhibited the lowest unoccupied molecular orbital (LUMO) of 1.49 eV and a highest occupied molecular orbital (HOMO) of −6.77 eV, forming a bandgap (*ϕ*_g_) of 8.26 eV, thus showing excellent electrical insulating characteristics. Oxidation lead to a lower LUMO of −1.92 eV with an increased HOMO of −5.68 eV. Meanwhile, charge trapping sites were formed near LUMO and HOMO, which exhibited *ϕ*_te_ = 3.56 eV for electrons and *ϕ*_th_ = 1.90 eV for holes. Apparently, the bandgap *ϕ*_g_ decreased to 3.76 eV, indicating substantially reduced insulating performance.

Most commonly in the power industry, copper electrodes are applied, exhibiting a work function of −4.7 eV. As the charge injection barrier (*ϕ*_inj_) is defined as the energy discrepancy between the work function of a metal electrode and the LUMO of a dielectric polymer for electrons, oxidation lead to tremendously decreased *ϕ*_inj_ (6.2 eV for LDPE and 2.8 eV for oxidized LDPE). Similarly, *ϕ*_inj_ for holes synchronously decreased from 2.1 eV to 1.0 eV.

The prolonged oxidation duration could have caused more grafted O atoms onto the LDPE, which may have further modulated the energy structure of the dielectric polymer. Here, the molecular structures of C_20_H_36_O_3_, C_20_H_32_O_5_ and C_20_H_28_O_7_ were formed, as demonstrated in Figure 5a, and the results are indicated in Figure 5b. The introduction of O atoms lead to a tremendously lowered LUMO below the vacuum level, and C_20_H_40_O, C_20_H_36_O_3_, C_20_H_32_O_5_ and C_20_H_28_O_7_ exhibited LUMO values of −1.92 eV, −1.94 eV, −1.99 eV and −2.08 eV, respectively. Therefore, a continuous surface oxidation process could result in a lower LUMO. Comparatively, the corresponding HOMO levels were −5.69 eV, −5.73 eV, −5.65 eV and −5.80 eV, respectively. With the variations in HOMO and LUMO levels, the oxidized LDPE exhibited bandgaps of 3.76 eV, 3.63 eV, 3.65 eV and 3.72 eV, respectively, which was far smaller than that of the as-prepared LDPE with 8.26 eV. However, the absolute quantity of introduced O atoms did not effectively influence the bandgap. In terms of the charge trapping site, with an increased number of introduced O atoms, the trap level for electrons decreased from 3.56 eV of C_20_H_40_O to 2.84 eV of C_20_H_28_O_7_, while the trap level for holes gradually increased from 1.90 eV to 2.86 eV.

A comparable simulation by T. Takada et al. suggests a LUMO energy level of 2.20 eV, a HOMO energy level of −7.69 eV and a bandgap 9.89 eV for C_24_H_50_ [45]. With one grafted O atom, the LUMO and HOMO energy levels of C_24_H_48_O changd to −0.91 eV and −6.84 eV, respectively, with the formed band gap of 5.93 eV, which is consistent with our results.

It should be noticed that the first-principle simulation applies the optimized molecular structure with minimized energy at the theoretical temperature of absolute zero. Therefore, the calculated energy structure might be different from practical conditions. For instance, the common acknowledgement of charge injection barriers is within 1.0–1.5 eV for a typical contact between a copper electrode and a dielectric polymer in practical applications, which is far smaller than the calculated 6.2 eV for electrons and 2.1 eV for holes in an LDPE/copper system. In fact, a 6.2 eV barrier height could lead to a blockage of space-charge injection, which is apparently unrealistic in real conditions.

As the quantity of space charges is directly controlled by a charge injection barrier, and charge injection barrier is modulated by the degree of surface oxidation (represented by the number of grafted O atoms), and a qualitative investigation can be achieved, as shown in Figure 6. In Figure 6, the as-prepared LDPE film exhibited a charge injection barrier of 2.12 eV based on molecular simulations, and short-term surface oxidation (C_20_H_40_O) resulted in a tremendously decreased charge injection barrier at about 1.03 eV. Further prolonging of the surface oxidation duration did not apparently affect the charge injection barrier, as it is maintained a range of 1.00–1.15 eV. The decreased charge injection barrier resulted in more charge injection and accumulation, as the space-charge quantity increased from 141 nC for as-prepared film to 469 nC of 1 h oxidized film. Such a large space-charge quantity was maintained by further enhancing the surface oxidation duration, the trend of which shows a perfect contrast with that of barrier height.

### 3.4. Space-Charge Distributions and Electric Field Distortion

Here, in order to systematically investigate the electrode/polymer interfacial barrier-dependent space-charge behaviors of the oxidized LDPE, numerical simulations were carried out, and the results are reported in Figure 7.

When determining the range of the charge injection barrier, several articles are referenced. A charge injection barrier was estimated by G. Chen’s group through an improved trapping/detrapping model, and they concluded that the charge injection barrier was in range of 1.1–1.3 eV for as-prepared dielectric films [46]. Similarly, the charge injection barrier was reported as 1.2–1.24 eV by simulations by Y. Zhou’s group [47]. As surface modification may greatly change the charge injection barrier, this work assumes a wider range of 1.1–1.5 eV, as we considered such a wide range could benefit scientific understanding and provide a data-set for space-charge behaviors related to charge injection barriers.

In the simulations, a 1–9 kV DC voltage was stressed onto a 100 μm thickness LDPE film for up to 20,000 s, and the electrode/polymer interfacial charge injection barrier was set as a variable in the range of 1.1–1.5 eV. A calculating grid of 200 was utilized, and thus the space charge distribution accuracy was controlled to 0.5 μm along the film depth direction. First, the total space charge (*N*_sp_) with the variations in the charge injection barrier was investigated, as reported in Figure 7.

Figure 7a applies a voltage of 1 kV (1 × 10^7^ V/m). At the charge injection barrier of 1.1 eV, space charges could be sufficiently injected under 10^7^ V/m. During the first few seconds, *N*_sp_ of ca. 10^−6^ C/m^2^ was formed, and rapidly increased towards 10^−4^ C/m^2^ in 500 s. After that, a balanced *N*_sp_ was gradually achieved until the end of 20,000 s. By increasing *ϕ*_inj_ by 0.1 eV, the initial *N*_sp_ dropped to 1.6 × 10^−8^ C/m^2^. With the prolonged voltage application duration, the increasing rate of space-charge quantity was similar to that under a 1.1 eV barrier height, showing a nearly parallel configuration in a log(total space charge)-log(duration) plot. *N*_sp_ continuously increased during the applied 20,000 s, thus, the balanced space-charge accumulation was not reached. Further increasing *ϕ*_inj_ lead to tremendously decreased space-charge density, and thus the soaring of space-charge quantities after oxidation was mainly contributed by the reduced charge injection barrier height, rather than the newly generated charge traps in the bulk of the material.

The applied voltage increased to 5 kV and 9 kV for further discussions of space-charge behaviors, as shown in Figure 7b,c, respectively. Apparently, increasing the applied voltage lead to enhanced space-charge injection. For *ϕ*_inj_ = 1.1 eV, the applied 5 kV voltage resulted in an initial *N*_sp_ of 3.2 × 10^−5^ C/m^2^, while that under 9 kV was 2.5 × 10^−4^ C/m^2^. The balanced *N*_sp_ also showed an increase with the increased applied voltage. An accelerated charge injection naturally lead to an earlier arrival of a balanced state. For instance, under *ϕ*_inj_ = 1.1 eV, it required 1000 s to reach the saturation state with space charges of 2.2 × 10^−4^ C/m^2^ for 1 kV voltage conditions, while those are 800 s with 1.2 × 10^−3^ C/m^2^ and 100 s with 1.8 × 10^−3^ C/m^2^ when 5 kV and 9 kV were applied to the LDPE film, sequentially. In Figure 7b,c, the saturation state could be observed for *ϕ*_nj_ ranging in 1.1–1.5 eV, however it only existed with *ϕ*_inj_ = 1.1 eV under 1 kV voltage. Figure 7d shows the applied voltage-dependent *N*_sp_ at 20,000 s, which directly indicates that the charge injection barrier dominated the space-charge accumulation.

Space-charge distributions greatly affected the inner electric filed distribution of the film, leading to a varied electrical breakdown performance. Here, the space-charge distributions of the LDPE with an electrode/dielectric interfacial barrier height of 1.1 eV and 1.5 eV, under 5 kV voltage, were compared, and the results are shown in Figure 8. Under *ϕ*_inj_ = 1.1 eV, large quantities of space charges were injected, and distributed in the vicinity of the left electrode. However, no space charge could be observed beyond 6 μm near the left electrode. With a prolonged applied voltage duration towards 1000 s, continuous injection occurred, and the space-charge density reached 56 C/m^3^. The strong electric stress promoted space-charge migration to the inner bulk of the film, as its density reached 10 C/m^3^ at a depth of ca. 50 μm. Space-charge reached the right electrode by 1000 s duration, and its density kept ca. 13 C/m^3^ with the prolonged duration.

As is discussed in Figure 7, the charge accumulation was tremendously suppressed with high interfacial barriers. In Figure 8b, apparently, densities of space charges were exponentially smaller than those under the conditions of *ϕ*_inj_ = 1.1 eV, as only 10^−4^ C/m^3^ was reached. By increasing the voltage application duration, the space-charge density in vicinity of the left electrode continuously increased, with observable space-charge migration towards the right electrode.

According to Poisson’s equation, less space-charge accumulation results in less distortion of the inner electric field along the film depth direction of LDPE [48], and the electrical field distributions corresponding to Figure 8a,b are shown in Figure 8c,d, respectively. As most of space charges were accumulated near the left electrode in Figure 8a, these homogeneous charges triggered an observable reduction in the electric field in the 10 μm vicinity of the electrode. This distortion became more violent with an enlarged accumulated space-charge density, and the drop of the electrical field even reached 1 × 10^7^ V/m. Electrons started to accumulate near the right electrode when the voltage application duration exceeded 100 s, and these formed heterogeneous charges that elevated the corresponding electric field. After 1000 s, the electric strength near the right electrode reached 7 × 10^7^ V/m, which exhibited a far-larger value of 5 × 10^7^ V/m. Thus, the possibility of breakdown increased near the right electrode, and an applied voltage with a value of 2 kV/mm below the theoretical breakdown strength of LDPE could trigger a breakdown. Apparently, in terms of electric field distortion, the accumulation of space charges did not benefit the electrical insulating performance. Thus, such space-charge injection should be strictly prohibited, and one efficacious strategy is to increase the interfacial barrier height, as indicated in Figure 8d. In Figure 8d, although the electric field distribution profile was similar to that in Figure 8c, showing a decrease near the left electrode alongside an increase near the right electrode, the absolute value of the distorted electric stress was inhibited to 10^2^ V/m. These results suggest that the space charge-induced electric field distortion could be systematically ignored when considering the issue of electrical breakdown.

As the TSC and PEA results in Figure 2 and Figure 3 suggest enhanced space-charge accumulation, and the energy structure results in Figure 4 and Figure 5 reveal decreased charge injection barrier, as theoretically, surface oxidation leads to the deterioration of breakdown performance, which is in accordance with earlier investigations.

In this work, the variations in the charge injection barrier by surface oxidation of polymeric dielectrics were investigated, as well as their influence on space-charge behaviors. Generally, surface oxidation lead to a decreased charge injection barrier, which promoted space charge injection and accumulation. This increased absolute space-charge amount enhanced the electric field distortion along the film depth direction, which lead to an enlarged inner electric strength (higher than the applied field), and consequently resulted in a decreased breakdown strength. Hence, when considering improving the electrical breakdown performance of a dielectric polymer, surface modification methods such as oxidation should be strictly forbidden. Oppositely, this fact leads to the confirmation of the validity of space-charge suppression methods such as surface fluorination, introduction of a dielectric barrier and interfacial deep-charge-trapping sites in improving breakdown performance.

In terms of surface dielectric performance, commonly, surface oxidation leads to an enlarged surface conductance, which is beneficial for inhibiting charge distribution concentration and surface electric field distortion. Consequently, the flashover performance of the dielectric polymer is improved. Generally, surface oxidation contributes to promoting a better flashover performance, but probably reducing the breakdown performance, showing a double-edged sword effect. Thus, the dielectric and electrical insulating performances can be well controlled and enhanced by selecting an appropriate surface modification method.

## 4. Conclusions

In this work, surface oxidation triggered variation in the electrode/dielectric charge injection barrier, and its function on the space-charge behaviors of LDPE films were investigated. Although multiplied quantities of bulk space charges were observed in the surface-oxidized LDPE, it was found that they were mainly induced by a decreased barrier height between the metal electrode and the dielectric polymer, which promoted space-charge injection. Further enhancing the surface oxidation duration lead to a lowered LUMO level and an elevated HOMO level of the polymer dielectrics, which consequently resulted in a decreased charge injection barrier for both electrons and holes. Through first-principle and numerical simulations, surface oxidation was found to modify the interfacial charge injection barrier to an extreme extent, and thus the accumulated space charges in the bulk of the dielectric polymers could differ in an exponential level, leading to a greatly varied electric field distortion along the film depth direction. By selecting an appropriate surface modification method in controlling electrode/dielectric interfacial barriers, space-charge accumulation could be selectively promoted or inhibited, which consequentially benefitted the flashover and electrical breakdown performance of the dielectric films.

## Figures and Tables

**Figure 1 materials-15-06095-f001:**
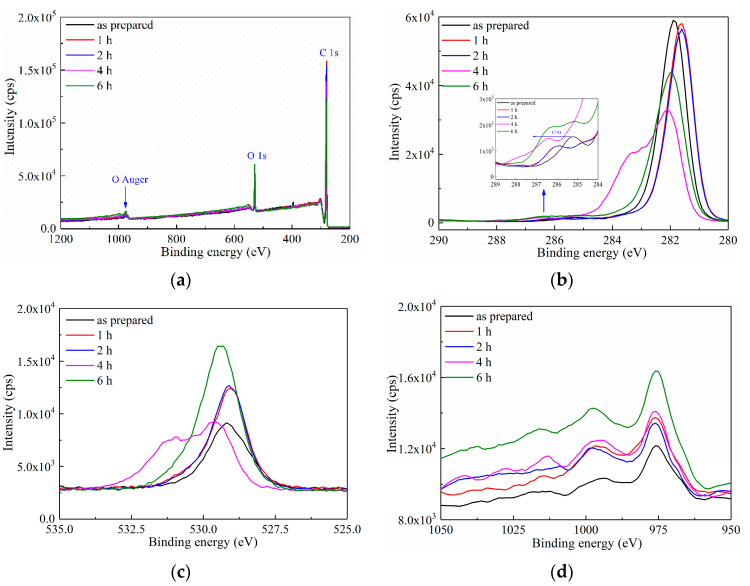
XPS spectra of as-prepared LDPE film and LDPE films with surface oxidation durations of 1–6 h. (**a**) XPS spectrum in binding energy range of 200–1200 eV; (**b**) The C 1s XPS spectrum; (**c**) The O 1s XPS spectrum; (**d**) The O Auger electron XPS spectrum.

**Figure 2 materials-15-06095-f002:**
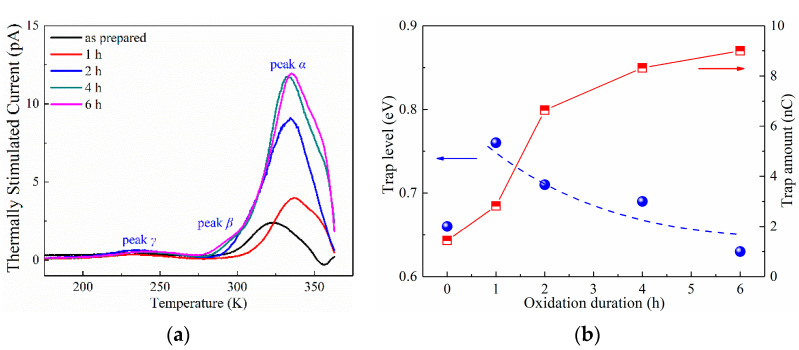
TSC spectra of as-prepared LDPE film and LDPE films with surface oxidation durations of 1–6 h. (**a**) TSC spectra in temperature range of 170–365 K; (**b**) The variations in trap level and trap amount with changes in surface oxidation duration, obtained from TSC results.

**Figure 3 materials-15-06095-f003:**
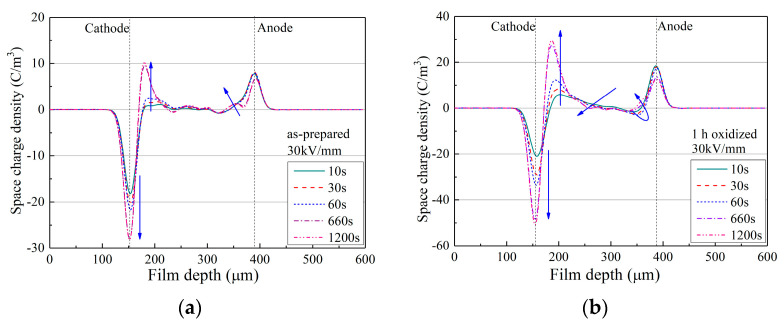
PEA results of as-prepared LDPE film and LDPE films with surface oxidation durations of 1–6 h. (**a**) PEA result of as-prepared LDPE film; (**b**) PEA result of LDPE film after 1 h surface oxidation; (**c**) PEA result of LDPE film after 6 h surface oxidation; (**d**) Comparison of space-charge accumulation of LDPE after surface oxidation, tested by ISPD, PEA and TSC methods.

**Figure 4 materials-15-06095-f004:**
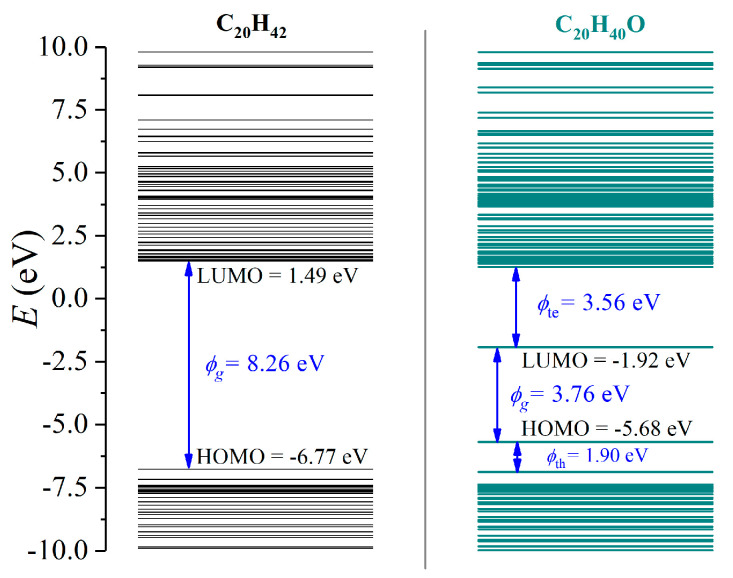
The calculated energy band structures of LDPE with a polymerization index of 10 (C_20_H_42_) and surface oxidation treated LDPE (C_20_H_40_O).

**Figure 5 materials-15-06095-f005:**
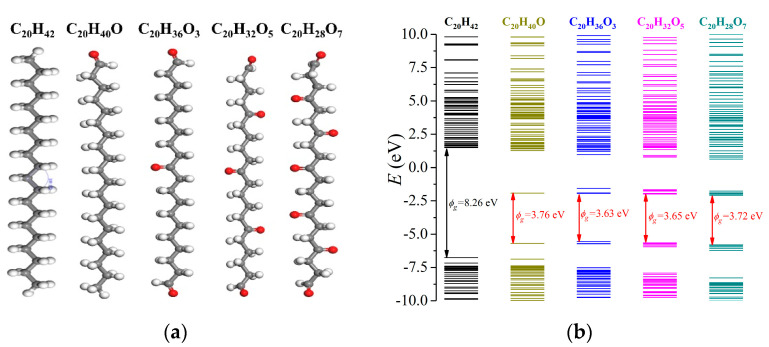
Variations in energy band structures of LDPE with surface oxidation treatment. (**a**) Molecular structures of surface oxidized LDPE with gradually increased grafted O atoms; (**b**) The calculated energy band structures of surface oxidized LDPE with gradually increased grafted O atoms.

**Figure 6 materials-15-06095-f006:**
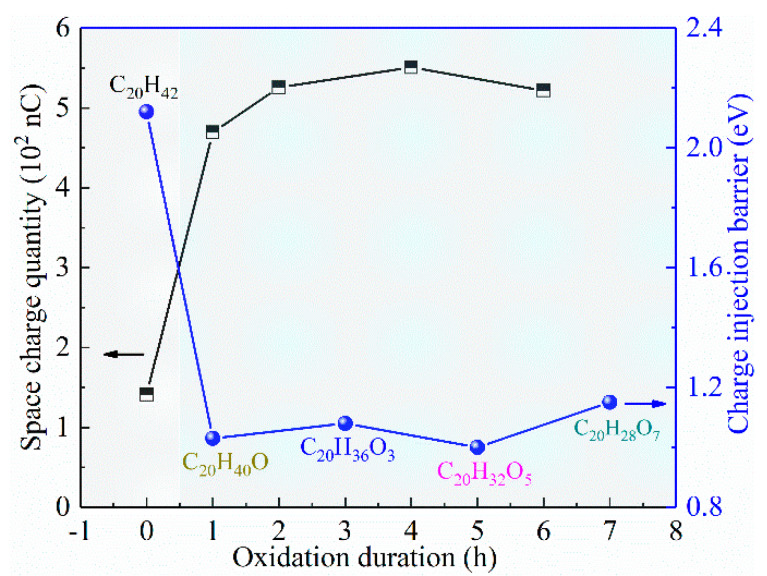
Correlations between space-charge quantity (obtained from PEA experiments) and charge injection barrier (obtained from molecular simulations).

**Figure 7 materials-15-06095-f007:**
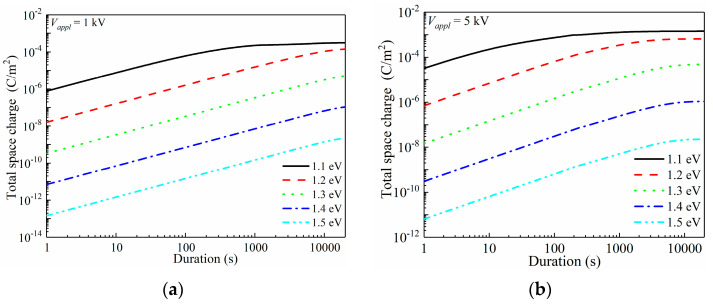
Electrode/polymer interfacial barrier-dependent space-charge accumulation characteristics of LDPE. (**a**) Total quantity of space charges in LDPE under 1 kV DC voltage (1 × 10^7^ V/m) with 1.1–1.5 eV charge injection barriers; (**b**) Total quantity of space charges in LDPE under 5 kV DC voltage (5 × 10^7^ V/m) with 1.1–1.5 eV charge injection barriers; (**c**) Total quantity of space charges in LDPE under 9 kV DC voltage (9 × 10^7^ V/m) with 1.1–1.5 eV charge injection barriers; (**d**) Changes in total quantity of space charges under variations in applied voltage (1–9 kV) with 1.1–1.5 eV charge injection barriers.

**Figure 8 materials-15-06095-f008:**
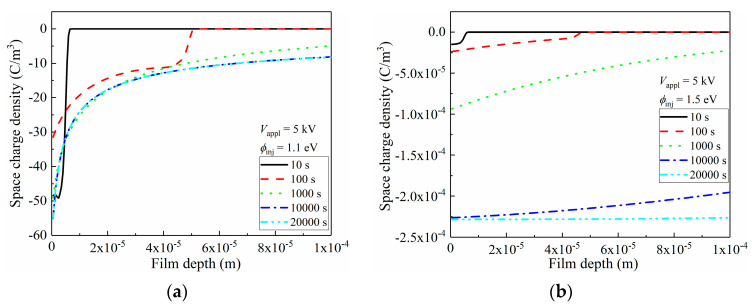
Space-charge distributions and induced electric field distortion inside LDPE film, under 5 kV (5 × 10^7^ V/m) applied voltage. (**a**) Distribution of space charges along film depth direction in LDPE with 1.1 eV charge injection barrier; (**b**) Distribution of space charges along film depth direction in LDPE with 1.5 eV charge injection barrier; (**c**) Distribution of electric field along film depth direction in LDPE with 1.1 eV charge injection barrier; (**d**) Distribution of electric field along film depth direction in LDPE with 1.5 eV charge injection barrier.

**Table 1 materials-15-06095-t001:** The obtained charge trap parameters of surface oxidized LDPE films.

Film	Trap Level/eV	Trap Amount/nC
LDPE	0.66	1.44
Oxidized by 1 h	0.76	2.82
Oxidized by 2 h	0.71	6.64
Oxidized by 4 h	0.69	8.32
Oxidized by 6 h	0.63	9.00

## Data Availability

The data presented in this study are available on request from the corresponding author.

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
