# Peer review of "Surface Modification-Dominated Space-Charge Behaviors of LDPE Films: A Role of Charge Injection Barriers"

_materials, 2022, doi:10.3390/ma15176095_

Round 1
Reviewer 1 Report
In this manuscript, the authors discuss and investigate the surface oxidation triggered variation of electrode/dielectric charge injection barrier, and its function on space charge behaviors of LDPE films. This is a helpful and interesting study with working principle and simulations.
Unfortunately, there is no experimental results to validate the simulation and show a more comprehensive result.
1. Please please add some experiments data to prove or validate the conclusion.
2. Please discuss more on the limitation or drawbacks of the mechanism.
Author Response
We thank the reviewers for their positive assessments of this work and helpful suggestions to improve our manuscript. In the following, our detailed response is itemized. Please see the attachment.

Reviewer 2 Report
No further comments.
Author Response

(The authors gave the same response as above.)

Reviewer 3 Report
In this work, the authors investigated the charge injection barrier, and its impact on the properties of the space charge in the surface oxidized LDPE film, a typical dielectric polymer.
The authors concluded that the electrode/polymer interfacial barrier, rather than the modification of the trap energy levels, plays an essential role in the observed space charge accumulation.
It was concluded that by appropriate choosing the surface modification method, potentially better performances regarding breakdown and flashover is expected for high-voltage applications.
Both experimental measurements and theoretical simulations have been performed.
XPS spectrum was utilized to infer the concentration of oxygen elements in the surface layer of LDPE films.
Also, information on LDPE film's space charge characteristics was extracted using TSC spectroscopy.
As the most prominent peak of the TSC spectrum is attributed to the accumulated space charges, the authors concluded that space charge accumulation becomes more significant as the duration of ozone treatment increases.
In particular, the trap level and trap amount are analyzed for different amounts of ozone treatment.
As the authors argued, the oxidation process is restricted mainly to several micrometers of polymer film and does not affect the bulk chemical structure.
The results of the TSC current, on the other hand, indicate that space charge injection is significantly affected by oxidation duration.
As a result, it leads to sizable change variation in the accumulated charges in the bulk of the film and, consequently, augmented alpha peak in the TSC spectrum.
Meanwhile, the trap levels are founded mostly unchanged when compared with the trap amount, which varies significantly, as shown in Fig.2.
This leads to the main conclusion claimed by the authors.
The authors' conclusion was further strengthened by the theoretical analysis of the charge injection barrier.
To this end, the energy level structures of C20H42, C20H42O, and other oxygen-grafted counterparts, up to C20H28O7, are evaluated using the ORCA package in conjunction with the B3LYP function.
Comparing the LUMO and HOMO of the above polymers showed that further oxidation facilitates the injection process.
Finally, space charge accumulation was evaluated as a function of oxidation duration, externally applied tension, and film depth.
Non-conjugated polymeric insulating materials play an essential role in power equipment applications, particularly for scenarios with significant transmission voltage.
As the device's performance depends on oxidation defects, proper experimental characterization of the material is very important.
Space charges injected from the metal electrode inevitably accumulated in the bulk of the insulator.
This process is the main reason for the accelerated deterioration and breakdown of the polymeric insulating materials.
In practice, the gradually increased power transmission voltage calls for improvement in the high-voltage capability of polymeric insulators.
Therefore, such studies are relevant from a practical perspective.
On the theoretical side, divergence still exists in the literature regarding the surface modification in polymer insulators, and in particular, the potential trade-off between the roles of the space charge and surface charge is a relevant topic.
The paper is mainly well-written, and the methodologies and arguments seem sound.
However, I have the following concerns before recommending its publication.
(1) The authors wrote
"Earlier investigation indicates ... and peak alpha is generated by accumulation of space charges."
and later
"As it is widely acknowledged, surface oxidation could only occur within several micrometers of polymer films, and the chemical structures in bulk of the film stays unchanged.
TSC represents the space charge behaviors of the film bulk, in other words, the quantity of space charges along the film depth direction is integrally accounted."
As the arguments and main conclusion of the manuscript critically reside on these statements, proper citations to the source(s) of the statement should be provided.
Experimentally, I am not entirely convinced by the claim, primarily drawn from the right plot of Fig.2, where the uncertainties of the estimations are not presented.
(2) The theoretical calculations indicate that the more significant change in the gap occurs between C20H42 and C20H40O.
Also, when considering the work function of the metal electrode, the most significant decrease in the injection barrier appears at the moment of a single grafted oxygen atom.
I wonder whether this theoretical speculation can be verified or rejected through some measurements.
Moreover, it is relevant to show how the above calculations of static barrier value related quantitatively to the dynamic oxidation process presented in the right plot of Fig.2.
Is this what is evaluated in Fig.5?
Can these results be (qualitatively) compared against/related to the right plot of Fig.2, particularly about the features shown there, by adopting proper parameters and transforming intensive quantities into extensive ones?
I suggest the authors elaborate further on these points.
Minor issues:
(1) Proper reference (regarding the assumptions and approximations that give rise to the formula) should be given when Eq. (1) is utilized to extract the trap level from the data.
(2) There are a few typos. For instance, in the sentence "As it is widely acknowledged, surface oxidation could only occur within several micrometers of polymer films, and the chemical structures in bulk of the film stays unchanged." "in bulk of" should be "in the bulk of," and the word "stays" should be "stay."
Author Response

(The authors gave the same response as above.)
